# Microstructural, Biomechanical, and In Vitro Studies of Ti-Nb-Zr Alloys Fabricated by Powder Metallurgy

**DOI:** 10.3390/ma16124240

**Published:** 2023-06-08

**Authors:** Eyyup Murat Karakurt, Yuksel Cetin, Alper Incesu, Huseyin Demirtas, Mehmet Kaya, Yasemin Yildizhan, Merve Tosun, Yan Huang

**Affiliations:** 1BCAST, Institute of Materials and Manufacturing, Brunel University London, Uxbridge, London UB8 3PH, UK; eyyupmurat.karakurt@brunel.ac.uk; 2TUBITAK, Marmara Research Center, Life Sciences, Medical Biotechnology Unit, Kocaeli 41470, Turkey; yasemin.yildizhan@tubitak.gov.tr (Y.Y.); mervetosun0@outlook.com (M.T.); 3TOBB Technical Sciences Vocational School, Karabuk University, Karabuk 78050, Turkey; alperincesu@karabuk.edu.tr (A.I.); hdemirtas@karabuk.edu.tr (H.D.); 4Machinery and Metal Technologies Departmant, Corlu Vocational School, Tekirdag Namik Kemal University, Tekirdag 59830, Turkey; mehmetkaya@nku.edu.tr

**Keywords:** powder metallurgy, porosity, space holder technique, corrosion resistance, biocompability

## Abstract

This study investigated the microstructures, mechanical performances, corrosion resistances, and in vitro studies of porous Ti-xNb-10Zr (x: 10 and 20; at. %) alloys. The alloys were fabricated by powder metallurgy with two categories of porosities, i.e., 21–25% and 50–56%, respectively. The space holder technique was employed to generate the high porosities. Microstructural analysis was performed by using various methods including scanning electron microscopy, energy dispersive spectroscopy, electron backscatter diffraction, and x-ray diffraction. Corrosion resistance was assessed via electrochemical polarisation tests, while mechanical behavior was determined by uniaxial compressive tests. In vitro studies, such as cell viability and proliferation, adhesion potential, and genotoxicity, were examined by performing an MTT assay, fibronectin adsorption, and plasmid-DNA interaction assay. Experimental results showed that the alloys had a dual-phase microstructure composed of finely dispersed acicular hcp α-Ti needles in the bcc β-Ti matrix. The ultimate compressive strength ranged from 1019 MPa to 767 MPa for alloys with 21–25% porosities and from 173 MPa to 78 MPa for alloys with 50–56% porosities. Noted that adding a space holder agent played a more critical role in the mechanical behaviors of the alloys compared to adding niobium. The pores were largely open and exhibited irregular shapes, with uniform size distribution, allowing for cell ingrowth. Histological analysis showed that the alloys studied met the biocompatibility criteria required for orthopaedic biomaterial use.

## 1. Introduction

Ease of fabrication, relatively low cost, and combined with superior mechanical performances, have provided metals a key role in implantology. In this regard, various metallic biomaterials (i.e., titanium-based alloys, cobalt-chromium alloys, and stainless steel) have been developed by scientists. However, a biomechanical mismatch between the host bone and the metallic biomaterial is one of the core issues of concern [1,2,3]. A majority of metallic biomaterials may exhibit much higher mechanical performances than natural bone. Such cases may lead to a reduction in bone mineral density or bone resorption over time, which is a known stress-shielding issue. This issue may increase healing time and even result in implant failure [4,5]. Therefore, the mechanical properties of the implant material should be close to those of the natural bone. However, using porous implant materials can be advantageous for implantology since their mechanical properties match better those of bone and thus can minimize the stress shielding issue. Porous implant materials are also essential for tissue engineering, as porous structures within an implant can promote adequate sites for bone ingrowth and physiological transport. Therefore, Ti-Nb-Zr alloys with different porosities were fabricated by powder metallurgy in the present study.

Ti and its alloys have lower density and superior biocompability and corrosion resistance compared to Co-Cr alloys, stainless steel, and other metallic biomaterials and thus, have been dominating the metallic biomaterials market for both orthopedic and dental applications [6]. There is an allotropic transformation in Ti-based alloys, and they are generally categorized based on their phase constituents, e.g., alpha (α), alpha + beta (α + β), and beta (β) type alloys [7,8]. Dual-phase α+ β Ti-6Al-4V alloy has been the mostly Ti-based alloy for biomedical applications as it possesses good workability and superior fatigue strength, in addition to an excellent specific strength and biological properties. However, the Ti-6Al-4V alloy was developed for aerospace engineering rather than for biomedical applications. The key alloying elements Al and V in the alloy may cause adverse influences in clinical practice. V ions released from Ti-6A-4V alloy may change the kinetics of the enzyme activity related to the inflammatory response [9,10]. Al may also lead to Alzheimer’s disease through long-term implantation [11]. This toxic issue must be addressed with the development of Al- and V-free Ti-based alloys. This is one of the motivations for the present work.

Ti-Nb-Zr alloys have recently been a focal point of interest in the development of advanced Ti-based implants. Both Nb and Zr are biocompatible and their addition to titanium allows the control of both phase constituents and microstructures, and therefore mechanical properties [12,13]. In particular, β stabilizer niobium has been used mainly for promoting the formation of ductile β phase [14]. Ti alloys with certain β phases have been generally preferred in implantology since they exhibit lower elastic modulus, more acceptable mechanical strength, and better biocompatibility than α Ti alloys [15]. Zr is a neutral or weak β-Ti stabilizer when individually alloyed to the Ti matrix [16]. Its addition is mainly for improving strength, fracture toughness and wear resistance [17]. In the Ti-Nb-Zr system to be developed in this study, the alloying element concentrations were designed to produce desired microstructures with both α and β phases [18]. Powder metallurgy was considered the best manufacturing technology for producing Ti alloys with controlled porosity, together with the space holder technology, and therefore selected. Powder metallurgy was also considered to be advantageous over casting in achieving homogeneous chemistry with the involvement of heavy metal Nb in the alloy. Careful consideration and optimization of the space holder agent are required to achieve the desired mechanical properties in the final product. According to previous studies, the quantity of space holder agents used in this study has been determined as 20% (wt.) [19].

This work aims to overcome the drawbacks of the above-mentioned problems by producing Ti-xNb-Zr10 (x: 10, and 20; at. %) alloys with differing porosities used as load-bearing implants that can mimic the bone structure. Microstructural analysis was performed by using various methods including scanning electron microscopy, energy dispersive spectroscopy, electron backscatter diffraction, and x-ray diffraction. Corrosion resistance was assessed via electrochemical polarisation tests, while mechanical behaviour was determined by uniaxial compressive tests. Biocompatibility was examined by performing an MTT assay, fibronectin adsorption, and plasmid-DNA interaction assay.

## 2. Materials and Methods

### 2.1. Materials and Alloy Fabrication

Ti (purity: 99.5%, particle size: 44 µm, Alfa Aesar, Haverhill, MA, USA), Nb (purity: 99%, particle size: 35 µm, Alfa Aesar), and Zr (purity: 99.7%, particle size: 23 µm, Alfa Aesar) powders were used in the fabrication of Ti-xNb-10Zr (x: 10 and 20; at. %) alloys having different porosities. Ammonium bicarbonate (NH_4_HCO_3_) (purity > 99%; Fisher, Hampton, NH, USA) was also used as a space holder (SH) agent to obtain the target porosities. Process parameters and chemical compositions are given in Table 1.

The powders of i, Nb, and Zr were mixed in a ball milling container, running at 24 rpm for 10 h to ensure compositional homogeneity. Each alloy was separately mixed with the addition of 20% (wt.) SH agent (NH_4_HCO_3_) to generate higher porosities. All powder mixtures were then uniaxially compacted under 300 MPa, in a tool-steel die with dimensions of Φ10 × 15 mm at ambient temperature. The constructive shape of the steel die tool used is given in Figure 1. Whereas the alloy samples with SH were heated to 180 °C and held for 2 h to remove SH from the bulk. Then, all alloy samples were sintered at 1200 °C for 6 h in an argon atmosphere, before being cooled down to 200 °C at a rate of approximately 5 °C/min. Consequently, the graph of the sintering regimes for the sample alloys with low and high porosities is given in Figure 2.

### 2.2. Metallographic Preparation, Characterization, and Mechanical Testing

For metallographic examinations, the sintered alloys were cut from different sections depending on observation needs. Then, the samples were ground by using abrasive papers up to 4000 grit and polished to 0.5 µm using a nano SiO_2_ suspension. The samples were etched chemically by using Keller’s solution (190 mL H_2_O, 5 mL HNO_3_, 3 mL HCl and 2 mL HF) for 8–10 s [20].

Microstructures were characterized with scanning electron microscopy (SEM) and energy diffraction spectroscopy (EDS). Electron backscattered diffraction (EBSD) analysis was conducted by the Zeiss Supra 35VP fitted with high sensitivity Digi View camera (EDAX Inc., NJ, USA). The working distance, the accelerating voltage, and the condenser aperture employed were 12 mm, 20 kV, and 120 μm, respectively, in high current mode. Furthermore, the step size selected was in a range of 0.2–1 μm according to the size of the area and grain size. Moreover, crystallographic characterizations of the sintered alloys with low and high porosities were performed by a Bruker D8 Advance Cu-Kα source. The step size, acquisition time, and the 2θ angle range used in this work were 0.02°, 1 s, and 30–80°, respectively.

The theoretical density, sintered density, and porosity of the alloys fabricated were calculated by using the following mathematical Equations (1)–(3) [21]. The theoretical density (𝜌_0_) was calculated from the atomic mass and specific densities of individual elements making up the alloys (Equation (1)). The sintered (bulk) density was directly determined from their mass-to-volume ratios (Equation (2)). The porosity was calculated by using theoretical density and sintered density (Equation (3)).
(1)$$\rho_{0}=\frac{M_{Ti}+M_{Nb}+M_{Zr}}{V_{Ti}+V_{Nb}+V_{Zr}}$$
(2)$$\rho =\frac{M}{V}$$
(3)ε=1−ρρ0*100
where ρ is the sintered (bulk) density (g/cm^3^), $\rho_{0}$ the theoretical density (g/cm^3^) and ε the porosity (%).

Mechanical testing was performed by uniaxial compression using samples of Φ10 mm × 15 mm. The compression tests were carried out three times for each sample condition on a universal tensile-compression testing machine (Zwick/Roell Z600) with a capacity of 100 kN, following EN 24506 standard. The average yield strengths and ultimate compressive strengths were determined from stress-strain curves (error range within 10%).

### 2.3. Electrochemical Analysis

The corrosion resistance performance of the material was analyzed by potentiodynamic polarization experiment on an E-Zastat potentiostat workstation, using an electrochemical flat cell that contains Hanks’ balanced salt solution (350 mL·cm^−2^, Fisher Scientific). The alloy sample, a standard calomel electrode, and a plate of platinum were employed as working electrodes (exposed area of 1 cm^2^), reference electrodes, and counter electrodes, respectively [22]. Potentiodynamic polarization curves were measured from −0.75 V (vs. open circuit potential) to 1.25 V with a scan rate of 0.01 V·s^−1^. Here, the corrosion potential (*E_corr_*) and the corrosion current density (*İ_corr_*) were determined by fitting the data to the Tafel curves. The polarization resistance (*R_p_*) was acquired by using the Stern Geary equation (see Equation (4)) [23].
(4)$$R_{p}=\frac{\beta_{a}\times \beta_{c}}{2303\times i_{corr}(\beta_{a}+\beta_{c})}$$
where *R_p_* is the polarisation resistance (Ω·cm^−2^), *βa* the Tafel slope of the anodic branch (V·dec^−1^), *βc* the Tafel slope of the cathodic branch (V·dec^−1^), and *i_corr_*: Corrosion density (µA·cm^−2^)

### 2.4. Biological Evaluations

The biocompatibility in vitro of sintered alloys with different porosities was investigated to evaluate cell viability and proliferation, cell morphology, adhesion potential, and plasmid-DNA interaction compared with reference TiGR4 material by following ISO 10993 [24].

Cell Culture conditions: Mouse fibroblast NCTC clone 929 cell line (L929), and human bone osteosarcoma cell line (Saos-2) supplied by American Tissue Culture Collection (ATCC, Manassas, VA, USA) were used for in vitro characterization studies. L929 and Saos-2 were grown in DMEM supplemented with 10% Fetal Bovine Serum (FBS, heat-inactivated, Gibco, #10500-064), and 1% Antibiotic-Antimycotic solution (10,000 units/mL of penicillin, 10,000 µg/mL of streptomycin, and 25 µg/mL of Amphotericin B, Gibco, Waltham, MA, USA, #15240-062) in 37 °C, 5% CO_2_ incubator. Their medium was changed with fresh DMEM every 2 days, and they were harvested with Trypsin-EDTA solution, 0.25% (Gibco, #25200-056) when they reached 70–80% confluency.

#### 2.4.1. Cell Viability Assays

MTT Assay: All alloy samples were sterilized at 120 °C by autoclaving for 30 min and extracted at a ratio of 0.2 g/mL in DMEM at 37 °C and 120 rpm for 72 h. L929 and Saos-2 cells were seeded in the range of 2.5 × 10^3^ to 1.5 × 10^4^ cells/well into the 96-well plate and incubated at 37 °C and 5% CO_2_. After 24 h, the 100 µL of extract from each sample was added to the cell monolayers and incubated at 37 °C and 5% CO_2_ for 1-day, 3-days, and 7-days. After each incubation period, 0.5 mg/mL of MTT (3-(4,5-Dimethyl-2-thiazolyl)-2,5-diphenyl-2H-tetrazolium bromide, Sigma-Aldhrich, St. Louis, MO, USA, #M5655)) in DMEM was added and then the cells were incubated at 37 °C and 5% CO_2_ for 4 h. After that, to dissolve the formazan crystals formed by viable cells, 100 µL of DMSO was added into each well and placed in a shaker for 2 h at room temperature. The absorbance of the formazan product was measured at 570–630 nm by a Microplate reader (Biotek Instruments, Inc., Santa Clara, CA, USA). The absorbance of the treated cells was normalized to the control cells and then cell viability was calculated as the percentage of the control.

Live-Dead Assay: The assay was performed as described in the previous study [25]. The cell viability test kit (Molecular Probes #L3224) is based on measuring cell viability-intracellular esterase activity with intensely green fluorescent calcein (ex/em ~495 nm/~515 nm) retained within the alive cells and plasma membrane integrity with Ethidium homodimer, EthD-1 (ex/em ~495 nm/~635 nm) binding to the nucleic acids in the membrane damaged dead cells. In parallel to the MTT cell viability assay, after each incubation period, the cells were washed twice with DPBS to remove serum esterase activity, and 100 μL of DPBS including 2 μM calcein AM and 4 μM EthD-1 was added to the cell monolayers and incubated for 30–45 min at room temperature. The images were taken under a fluorescence microscope (Leica DMI 6000) using 10× magnification.

#### 2.4.2. Cell Adhesion—Fibronectin Activity

Fibronectin (FN) adsorption was evaluated by a modified enzyme-linked immunosorbent assay (ELISA) method. The alloy disks and the TiGR4 reference disks were placed into the 24-well ELISA plates and coated with 250 µL of 1% Bovine serum albumin (BSA in PBS) and then incubated at 37 °C for 2 h. Following that, FN solution (2.5 µg/mL in PBS, Sigma, St. Louis, MO, USA) in 550 µL was added to the alloy disks and incubated for 2 h. Then washings with PBS, 550 µL primer monoclonal anti-fibronectin antibody (dilution 1:10,000, Sigma) was added into each well and incubated at 4 °C. After overnight incubation, the secondary antibody (goat anti-mouse immunoglobulin G conjugated with horse radish peroxidase, dilution 1:30,000, Sigma) was added and incubated at 37 °C for 30 min and then 550 µL of stop solution, 3.3′,5.5′-tetramethylbenzidine in 200 µg/mL (TMB for ELISA, Sigma) was added to the wells. When the formation of blue color was observed, 275 µL 2 M H_2_SO_4_ was added to the wells. Colorimetric detection was performed by reading the absorbance of the color intensity measured spectrophotometrically at 450 nm. The experiments were repeated in triplicate.

#### 2.4.3. Alloy Discs—Plasmid DNA Interactions

The pBOS-H2B-GFP plasmid (5.8 kb, BD Pharmingen William Saunders, San Jose, CA, USA) used for the assay was grown in E-coli and purified using a Machery Nagel DNA isolation kit. The alloy disks and the TiGR4 reference discs’ extracts were incubated with 200 ng of plasmid DNA in ddH_2_O for 16 h at room temperature in a total reaction volume of 20 µL. The samples and controls were electrophoresed on 1% agarose gels at 100 V for 1 h using TAE buffer. The gel was stained using ethidium bromide, and the images of the bands were taken by using the ChemiDoc imaging system (BioRad, Hercules, CA, USA).

#### 2.4.4. Cell Morphology—Scanning Electron Microscope

L929 and Saos-2 cells were seeded on disks and the TiGR4 reference discs (8 mm diameter, 0.2 mm thickness) at the range of 1 × 10^5^ to 4 × 10^5^ cells/well into the 24-well plate and incubated at 37 °C and 5% CO_2_ for 1-day and 7-day. After washing with PBS, the cells were fixed in the buffered 2.5% glutaraldehyde solution at room temperature for 1 h. Then, they were dehydrated using a series of ascending ethanol concentrations (60%, 70%, 80%, 90%, and 100% at 20 min each) and air-dried at room temperature for 24 h. Finally, a Field Emission Gun–scanning electron microscope (FEG-SEM), (QUANTA FEG 250) was used to examine the L929 and Saos-2 cells morphology on the alloy discs.

Statistical Analysis: It was performed using GraphPad Prism 8.0.2 and the statistical differences were calculated using one-way ANOVA and two-way ANOVA Tukey’s multiple comparisons test. The significance of each data point (n = 3) was determined when *p* < 0.05 was considered statistically significant. Interventional studies involving animals or humans, and other studies that require ethical approval, must list the authority that provided approval and the corresponding ethical approval code.

## 3. Results

### 3.1. Microstructure and Phase Composition

The porosity was analyzed and calculated for each individual alloy by using the theoretical and bulk density. The porosity results with error ranges are given in Figure 3. All sintered alloys exhibited two general porosity categories, low porosities of 21–25% for alloys fabricated without using the SH agent and high porosities of 50–56% for alloys made with the SH agent, respectively. This result was expected and suggested that the use of NH_4_HCO_3_ as an SH agent was effective in generating high porosities. No residues of NH_4_HCO_3_ were found in the alloys after sintering, which indicates that the agent was completely degraded to ammonia (NH_3_(g)) and carbon dioxide (CO_2_(g)) throughout the sintering stage, leaving pores in the material. The variations of porosity within both low and high categories were limited, suggesting that the processing parameters for mixing, compaction, and sintering were well-controlled and consistent. Having examined the effect of Nb addition on the porosity measured, it seemed that Nb concentration contributed, partially, to the variations of the porosity as the upper bound of porosities for both low and high porosity categories are associated with a high Nb concentration of 20%. This is because Ti and Nb have different diffusion coefficients at that temperature (1200 °C). This was consistent with a previous study, which showed that the porosity in Ti-Nb alloys sintered at 1500 °C increased with Nb concentration [26]. It should be noted that the porosities obtained in this work were within the range of porosities of human bones, which are between 10% and 85%, depending upon the type and age of the bone [27]. It is reasonable to believe that a full range of porosities required for clinical applications can be achieved by adjusting SH agent additions and sintering conditions, etc.

X-ray diffraction spectra are presented in Figure 4. XRD results revealed the coexistence of both hcp α (ICDD PDF No. 00-044-1294) and bcc β (ICDD PDF No. 00-044-1288) peaks related to the Ti-based alloys as well as primary Nb phase, but at different volume fractions and no other obvious impurities peak such as oxide, hydride or intermetallic compound was detected. Similarly, Sheremetyev et al. stated that α and β phases were found in the microstructure of Ti-14Nb-18Zr (at. %) alloys fabricated by the arc melting method [28]. In addition, alloying with Nb (strong β stabilizer) and zirconium (weak β stabilizer or neutral element) reduced the proportion of α phase in porous Ti-20Nb-10Zr (at. %) alloys with low and high porosities. Whereas β phase proportion increased by adding Nb, as Nb acted β-Ti stabilizer in the microstructure. On the other hand, adding a space holder decreased the intensities of solute diffraction peaks of both phases due to an increase in diffusion pathways between particles.

EDS analysis showing SEM micrographs and individual EDS points for Ti-10Nb-10Zr and Ti-20Nb-10Zr alloys with low porosities is depicted in Figure 5. The atomic compositions for individual EDS points are listed in Table 2. Different Ti, Nb, and Zr concentrations were detected across the examined areas of the alloys. The dark gray areas corresponding to the α phase were rich in Ti and Zr; white-gray areas corresponding to the β phase were rich in Nb; bright white areas corresponded to the primary Nb phase, although there were no undissolved Zr particles in the alloys studied. This was expected as Zr is fully dissolvable in the Ti matrix, but Nb has limited solubility in the α phase [29]. Therefore, in-homogeneous microstructures were obtained in all alloys due to the insufficient sintering temperature selected in this study.

SEM micrographs for all-sintered alloys with low and high porosities are shown in Figure 6. SEM micrographs showed that irregularly shaped pores with sharp corners were observed in the microstructures. The closed pore structures were determined in the alloys with low porosities, whilst open pore (interconnected) structures were found in alloys with high porosities, allowing for cell ingrowth and body liquid. SEM micrographs of the sintered alloys showing the pore morphology and distribution were evaluated to estimate the average pore size. In this regard, average pore sizes of the sintered alloys increased from 67 μm to 117 μm for Ti-10Nb-10Zr and 82 μm to 143 μm for Ti-20Nb-10Zr, as a consequence of the solid-state sintering. Accordingly, mean pore sizes and distributions for the alloys with high porosities were much higher than those for the alloys with low porosities, which revealed that the characteristics of pores achieved were consistent with the porosity level of the alloys. Two different types of pores were observed in the alloys achieved; these are macro and micropores. Micropores found in the alloys with low porosity were formed as a result of the sintering procedure, while the macropore observed in the alloys with high porosity was achieved as a result of the removal of space holders.

In addition, a fine acicular α phase transformation was found in Ti-10Nb-10Zr alloys, whilst an extremely fine acicular α phase was observed in Ti-20Nb-10Zr alloys due to increasing Nb concentration in Ti-Zr mixtures. In Ti-10Nb-10Zr and Ti-20Nb-10Zr alloys with low and high porosities, some areas with undissolved Nb cores surrounded by β phase were detected. Based on the literature reports, sintering is based on the physical diffusion phenomenon [30]. Therefore, in this work, the proportion of α/ β or dissolution of Nb cores in the Ti matrix were closely related to the sintering conditions used. Accordingly, the existence of Nb cores in the microstructures was an indication of insufficient sintering temperature. Rao et al. stated that higher sintering temperature affected the Nb dissolution in the Ti matrix, suppressing Widmanstatten morphology and increasing the volume fraction of the β phase [31]. A similar result was observed in this study. Here, an increase in Nb content from 10 (at. %) to 20 (at. %) increased the β phase formation and caused a finer acicular α phase transformation in the microstructure.

The EBSD technique was performed for a more detailed analysis of phase identification. The grain boundaries map and phase map for the Ti-10Nb-10Zr and Ti-20Nb-10Zr alloys with low porosities are shown in Figure 7. The findings depicted in Figure 7 confirmed the existence of both phases (α and β phases). Dual-phase microstructures consisting of bcc β grains and thin hcp α needles were observed. Nb and Zr concentrations in the titanium matrix reduced the martensite transition temperature (Ms) of the alloys, separately. Therefore, hcp α grains could not grow any further. The mean grain sizes obtained from the EBSD–IPF image for the hcp α phase in the Ti-10Nb-10Zr and Ti-20Nb-10Zr alloys with low porosities were calculated as 9.89 µm and 9.13 µm, respectively. Such conditions revealed that grain sizes for the α phase reduced with increasing Nb concentration, which were visible from SEM micrographs. Moreover, the proportion of hcp α phase in Ti-10Nb-10Zr alloy was greater than that in Ti-20Nb-10Zr alloy since Nb acted β-Ti phase stabilizing, which were 78% and 52%, respectively. Whereas, the volume fraction of the bcc β phase in alloys were detected as 11% and 23%, respectively. Lastly, undissolved Nb particles were surrounded by the β phase. Therefore, it was difficult to measure the size of the undissolved Nb cores. However, the EBSD map revealed that the proportion of the primary Nb phase in Ti-10Nb-10Zr alloy was smaller than that in Ti-20Nb-10Zr alloy, which were 11% and 25%, respectively.

### 3.2. Compressive Behaviour

Figure 8 presents typical stress-strain curves for all the sintered alloys showing differences in mechanical performances among individual alloys. The average yield strengths for the alloys with low porosities was 523 MPa for 10% Nb and 394 MPa for 20% Nb, while these were 155 MPa for 10%Nb and 63 MPa for 20%Nb for the high porosity alloys. The average ultimate compressive strengths for Ti-xNb-10Zr (x: 10 and 20; at. %)alloys with low porosities decreased from 1019 MPa to 767 MPa, whilst these for the same alloys with high porosities reduced from 173 MPa to 78 MPa, respectively. As foreseen, the mechanical performances of the alloys with high porosities were dramatically reduced. This could be mainly due to the porosity properties such as porosity level, pore distribution, and pore size influenced the crack initiation during compression tests, and thus cracks preferentially occurred in the expanded pores with increasing compressive strength, which caused serious decreases in mechanical performances of the alloys with high porosities [32,33,34]. Such behavior allows the implant to have a closer match to the mechanical properties of natural bone, minimizing stress shielding and facilitating improved load transfer between the implant and the bone tissue. When comparing the mechanical properties of stiff Ti-6Al-4V alloy with the sintered alloys achieved in this study, it can be said that the sintered alloys with low and high porosities can provide better mechanical compatibility with natural bone due to their reduced mechanical performances, lightweight, and potentially enhanced biocompatibility. These findings revealed that the sintered alloys with low and high porosities could be a favorable candidate biomaterial for implantology.

However, the mechanical properties of all sintered alloys achieved might be linked not only with porosity level but also with phase constituents (β/α phases). Accordingly, increasing Nb concentration from 10 (at. %) to 20 (at. %) decreased the mechanical performances, as Nb acted as β phase stabilizer. Previous studies stated that the mechanical properties of this type of Ti-phase were lower than those of the α phase. According to this concept, Ti-20Nb-10Zr alloy with low porosity exhibited higher β phase formation compared to Ti-10Nb-10Zr alloy with low porosity, resulting in lower UCS and yield strength values. Similarly, the same was observed in the alloys with high porosities. However, noted that adding a space holder agent did play a more critical role in the reduced mechanical performances of the sintered alloys comparing the α/β phase ratio.

### 3.3. Potentiodynamic Polarization

The typical cyclic potentiodynamic polarization (Tafel)curves for Ti-xNb-10Zr alloys with low and high porosities are given in Figure 9. The summary of the results is also given in Table 3. E_corr_ for Ti-xNb-10Zr (x: 10 and 20; at. %) alloys with low porosities were found in a range of −0.07 mV to −0.077 mV, respectively, in Hank’s balanced salt solution. At this point, the Tafel curve of Ti-20Nb-10Zr alloy with low porosity was slightly closer to the positive side of the corrosion potential, which revealed that this type of alloy had slightly lower current density (icorr: 2.26 µA·cm^−2^) compared to Ti-10Nb-10Zr alloy with low porosity (icorr: 2.40 µA·cm^−2^). As understood from these findings, an increase in Nb concentration from 10 to 20 slightly enhanced the corrosion resistance of Ti-20Nb-10Zr alloy with low porosity. The same tendency was observed in Ti-20Nb-10Zr with high porosity. This behavior showed that adding niobium and zirconium was attributed to the formation of an oxide film. The potential caused an increase in anodic current because the increase in oxide thickness during testing was not sufficient to compensate for the potential increase [14]. As a result, the alloys studied in this thesis were potential candidates for biomedical applications.

### 3.4. Biological Evaluations of Alloys

MTT Cell Viability Assay: L929 and Saos-2 cell lines exposed to all sintered alloys, and the reference TiGR4 extracts for 1-day, 3-day, and 7-day were reported in Figure 10. Based on the results achieved from the cytotoxicity assessment, the alloys achieved did not lead to cytotoxic effects and exhibited good viability of L929 and Saos-2 cell types for all incubation times. Further, they provided the criteria of biocompatibility required for the use of orthopaedic biomaterial (at least 70%) according to ISO 10993-5.

Cell viabilities upon exposure to the extracts of the examined alloys were found to be 77–97% for L929 and 72–80% for Saos-2 cell lines, which showed that alloys had significant differences in cell viability (*p* < 0.05). Here, the highest cell viability of L929 was determined in Ti-10Nb-10Zr alloy with 21% porosity as 97.02%, whereas the lowest one was observed in Ti-10Nb-10Zr alloy with 50% porosity, which was 76.96%. On the other hand, the maximum cell viability level for Saos-2 was found in Ti-20Nb-10Zr alloy with 25% porosity as 79.89%, whilst minimum one was observed in Ti-20Nb-10Zr alloy with 56% porosity, which was 72.48%. Results revealed that increasing Nb concentration from 10 (at. %) to 20 (at. %) in Ti-Zr mixtures resulted in a decrease in L929 cell viability and an improvement in Saos-2 cell viability. The treatment of Ti-10Nb-10Zr alloy with high porosity compared with low porosity one decreased the cell viability of the L929 cell line on the 1-day and 3-day but not on the 7-day application. This could be due to the adaptation of the cells to the conditions that occurred with high porosity samples and then the cell viability reached the same level, resulting in low porosity. The increase in Nb concentration from 10 (at. %) to 20 (at. %) in Ti-Zr alloys resulted in no significant change in L929 cell viability among the different periods of the treatments. The increase in Nb concentration from 10 (at. %) to 20 (at. %) in Ti-Zr alloys resulted in an increase in Saos-2 cell viability within the different periods of the treatments. It could be explained that the Saos-2 cell line might have a sensitive metabolic activity to the Nb metals, which may induce cell proliferation. In addition to these, Nb concentration might change the surface characteristics of the alloys by forming NbxO films.

The cell viability upon exposure to the alloy extracts was also confirmed by performing a live-dead assay and the images were presented in Figure 11. Green fluorescent calcein (ex/em ~495 nm/~515 nm) retained within the alive cells and red fluorescent Ethidium homodimer, EthD-1 (ex/em ~495 nm/~635 nm) bound to the nucleic acids in the membrane damaged dead cells. As seen in Figure 11, cell viability and proliferation of L929 and Saos-2 after 1-day and 7-day incubation was also qualitatively demonstrated with intensive green fluorescence appearance. All of the tested alloys were found to be biocompatible as compared with the control, DMEM, and reference material TiGR4. In conclusion, all incubation times showed good L929 and Saos-2 cell viabilities, which was a good indication for implantology and the alloys produced in this study were suitable for use as an orthopedic biomaterial.

The plasmid-DNA interaction of the alloy extracts was examined, and the result confirmed that no induced genotoxic potential was found in Appendix A. Plasmid DNA–Ti-(x)Nb-10Zr (x: 10 and 20; at. %) alloy interaction assay also demonstrated good biocompatibility of the developed alloys. None of the tested alloys did cause any DNA fragmentation, which proved the non-genotoxicity of the alloys presented by S (1).

Morphology of cells–Scanning Electron Microscope: To investigate the cell morphology, L929, and Saos-2 cell lines were seeded on all sintered alloy discs, and the cell images were taken with SEM on the 1-day and 7-day (Figure 12). The alloys achieved promoted adequate sites for biological fixation and cell ingrowth. Therefore, L929 and Saos-2 cell lines were well spread on the alloys achieved. In conclusion, multiple colonies of L929 and Saos-2 cell lines adhered to the alloys were found. The cell proliferation and adherence were increased towards the high porous region with the SH agent as seen in Figure 9.

Fibronectin adsorption: Protein adsorption occurs soon after the implantation of biomaterial within a biological environment and is a key determinant of the responses of cells to the material surface. L929 and Saos-2 cell proliferation, migration, and tissue integration were examined by measuring fibronectin protein adsorption with ELISA (Figure 13). The results demonstrated that the addition of SH agents and increasing the Nb concentration from 10 (at. %) to 20 (at. %) enhanced the fibronectin adsorption capacity. Further, Ti-20Nb-10Zr (at. %) alloy with high porosities exhibited even better fibronectin absorption rather than the reference material (*p* < 0.05). The fibronectin adsorption capacities of the alloys achieved were also correlated with the results of the cell viability and morphological images. Regarding the biological characteristics of Nb, it is nontoxic and allergy-free metal, indicating acceptable biocompatibility and osteo-conductivity. In this study, the increased level of Nb might induce much more mitochondrial activity, cell proliferation, and fibroblast adsorption potential of biomaterial surfaces. The previous work resulted in similar findings, that the comparative study on the biological performance of Nb, Ti, and stainless steel revealed that there were much more mitochondrial activity and cell proliferation on Nb compared to the others [35].

## 4. Discussion

This result was expected and suggested that the use of NH_4_HCO_3_ as a space holder agent was effective in generating high porosities. No residues of NH_4_HCO_3_ were found in the alloys studied after the sintering stage, which indicated that the space holder agent was completely degraded to ammonia (NH_3_(g)) and carbon-dioxide (CO_2(g)_), leaving pores in the material [36,37]. Accordingly, the alloys achieved in this study were fabricated by powder metallurgy with two categories of porosities, i.e., 20–25% and 50–56%, respectively. It is reasonable to believe that a full range of porosities required for clinical applications can be achieved by adjusting space holder agent additions and sintering conditions.

In implantology, the most critical point regarding a porous implant material is the size and distribution of macro and micropores in the microstructure. We can see conflicting research on the optimal pore size for implantology in the literature. However, most studies have shown that the optimal pore size should be in the range of 100 µm to 600 µm [36]. It is obvious that the pores within implant material should be larger than the minimum pore size allowing the blood and nutrient flow for bone ingrowth. Based on the results from the general porosity evaluations showed that alloys could be divided into low and highly porous categories, with pore sizes from 67 µm to 82 µm and 117 µm to 143 µm, respectively. Further, pore size and distribution found in all-porous alloys fabricated in this thesis were congruent with the pore size and distribution aimed for when the process began. However, the porosity level of load-bearing implant materials can vary according to where they are used in the body. Therefore, it is quite difficult to say which alloy has the optimal porosity level. Noted that all-porosity levels achieved in this study were appropriate for mimicking the bone structure, which positively affected the biocompatibility properties of the alloys. The outcome of this study showed that the alloys obtained could be considered a promising candidate for orthopedic applications.

In addition, the porosity characteristics could be effectively adjusted by adding an SH agent Meanwhile, adding Nb concentration into Ti-Zr mixtures changed α/β ratio in the microstructure and formation of β, which is of vital importance for biomedical applications due to unique mechanical properties, could be increased despite of insufficient sintering conditions. Thus, phase constituents were able to be modified to achieve appropriate microstructures.

Potentiodynamic polarization results showed that the Ti-xNb-10Zr alloys with low porosities met the corrosion performance criteria as required for orthopaedic biomaterial use although these alloys had inhomogeneous structures caused by powder metallurgy. This is mainly due to the individual elements making up the alloys being corrosion-resistant. Similarly, the same tendency was observed in the alloys with high porosities.

Previous studies revealed general porosities obtained were appropriate for reducing the risk of the stress shielding effect [38,39]. The alloys with high porosities exhibited mechanical performances closer to the human bone than the alloys with low porosities. Such conditions showed that powder metallurgy with the space holder technique was a crucial factor for adjusting the mechanical properties of the alloys studied in this study by generating extra porosity inside the microstructure.

Porosities obtained in this study provided adequate sites for bone ingrowth and cell proliferation [40]. However, SEM micrographs showing the morphology of the cells showed that better cell adhesion was observed in alloys with high porosities due to the interconnected open pore structures. In addition, Nb and Zr used as alloyant elements in the Ti matrix did not have any toxic or allergic effects on cell viability. Such conditions revealed that Al- and V-free Ti-based alloys could be fabricated by powder metallurgy method combined with the space holder technique [41,42]. This work was able to minimize the above-mentioned problems by producing Ti-xNb-10Zr (x: 10, and 20; at. %) alloys with different porosities used as load-bearing implants that can mimic the bone structure.

In summary, our findings on the microstructural, mechanical, and biological characterization of fabricated all-sintered alloys were supported by previous studies.

## 5. Conclusions

This study focused on the effect of Nb concentration on the microstructure, corrosion resistance, and mechanical performance of the alloys with differing porosities. Based on the results obtained, the following facts were determined. The powder metallurgy method combined with the SH technique was appropriate to produce Ti-xNb-10Zr (x: 10, and 20; at. %) alloys with differing porosities. The addition of the SH agent effectively adjusted the porosity characteristics of the alloys achieved. As foreseen, porosities dramatically diminished the mechanical performances of the alloys achieved in this study.

In this study, all sintered alloys were also characterized biologically following ISO 10993. The results demonstrated the good biocompatibility of tested alloys examining cell viability and proliferation, cell morphology, genotoxicity, and cell adherence potentials. The cell viability of used cell lines was increased with increasing incubation time. This result could be concluded that the fabricated alloys might be good candidates as implant materials, which need to stay in the body for a long time. Adding Nb concentration was not increased the biocompatibility levels, whereas adding an SH agent increased cell proliferation and fibronectin adsorption potentials of the alloys. In the SEM analysis, both cell lines moved towards the porous regions and highly proliferated in these regions.

## Figures and Tables

**Figure 1 materials-16-04240-f001:**
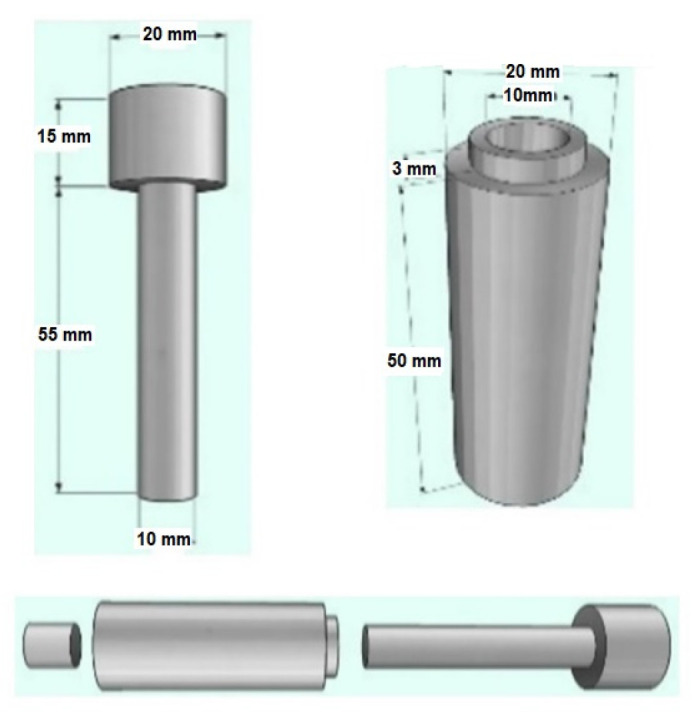
The constructive shape of the steel die tool used in the experiment.

**Figure 2 materials-16-04240-f002:**
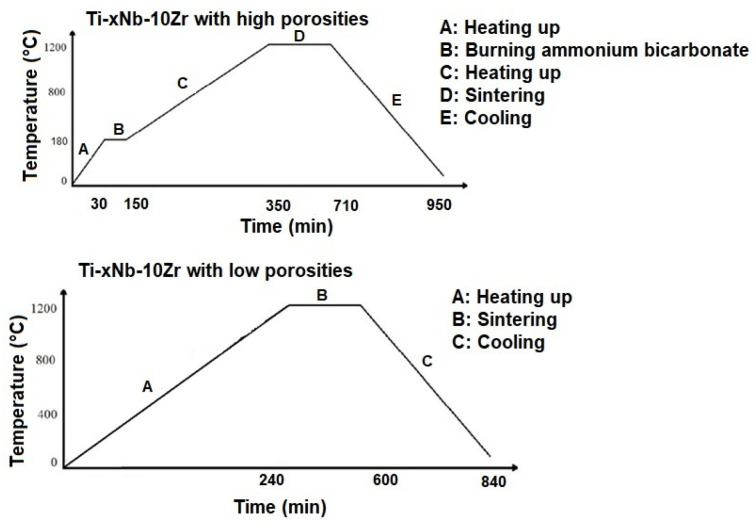
The sintering regimes for Ti-xNb-10Zr with low and high porosities.

**Figure 3 materials-16-04240-f003:**
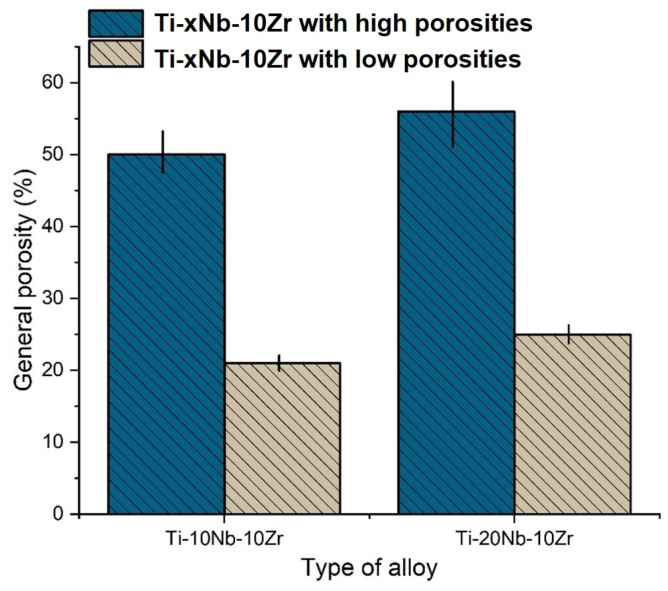
General porosity results of the alloys.

**Figure 4 materials-16-04240-f004:**
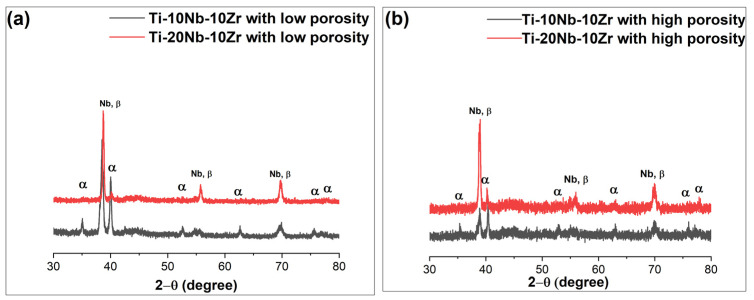
X-ray diffraction spectra for (**a**) Ti-xNb-10Zr (x: 10 and 20; at. %) with low porosities; (**b**) Ti-xNb-10Zr (x: 10 and 20; at. %) with high porosities.

**Figure 5 materials-16-04240-f005:**
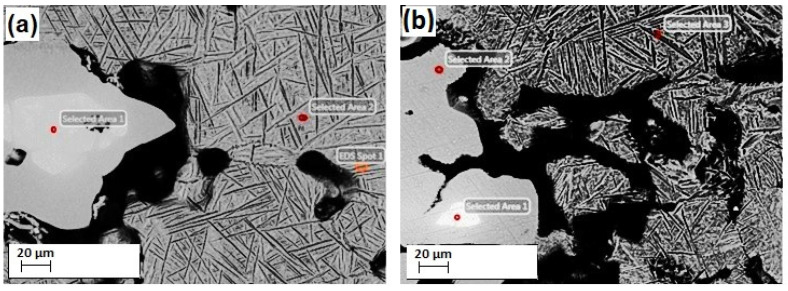
EDS analysis showing SEM micrograph and EDS points for (**a**) Ti-10Nb-10Zr with low porosity; (**b**) Ti-20Nb-10Zr with low porosity.

**Figure 6 materials-16-04240-f006:**
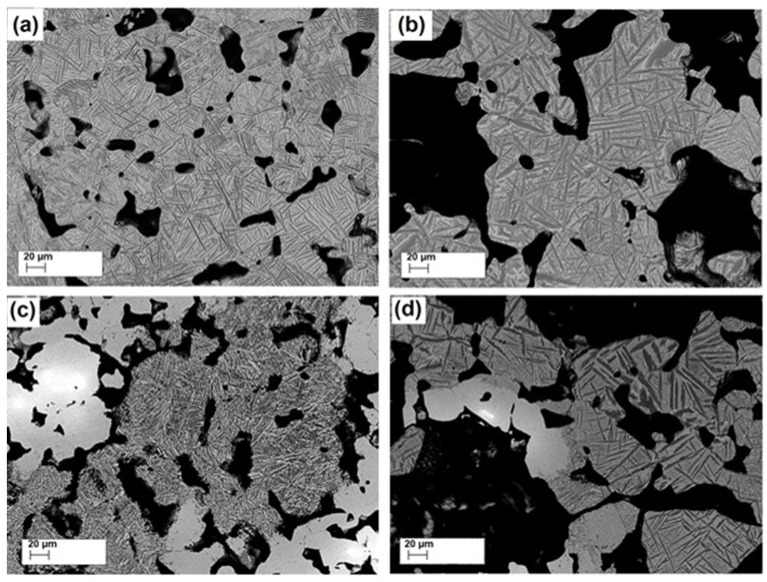
SEM micrographs for (**a**)Ti-10Nb-10Zr with low porosity, (**b**) Ti-10Nb-10Zr with high porosity, (**c**) Ti-20Nb-10Zr with low porosity, and (**d**) Ti-20Nb-10Zr with high porosity.

**Figure 7 materials-16-04240-f007:**
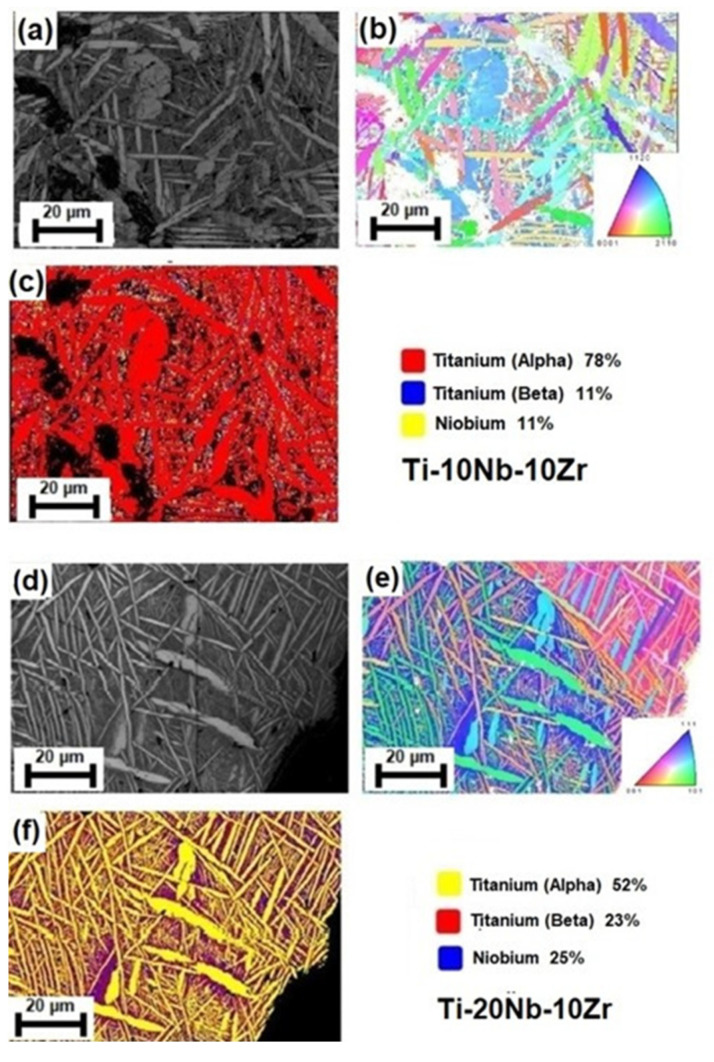
EBSD data showing the microstructure (IQ map), grain morphology (IPF map), and phase constituents and distribution (Phase map) for both Ti-10Nb-10Zr alloy and Ti-20Nb-10Zr alloy: (**a**,**d**) IQ map; (**b**,**e**) IPF map; (**c**,**f**) Phase map.

**Figure 8 materials-16-04240-f008:**
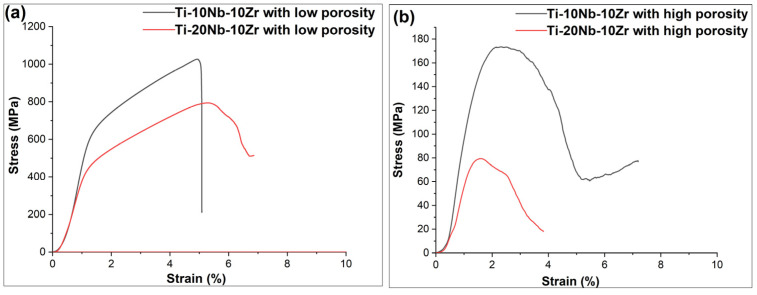
The Stress-strain curves for Ti-xNb-10Zr (x: 10 and 20; at. %) (**a**) alloys with low porosities, (**b**) alloys with high porosities.

**Figure 9 materials-16-04240-f009:**
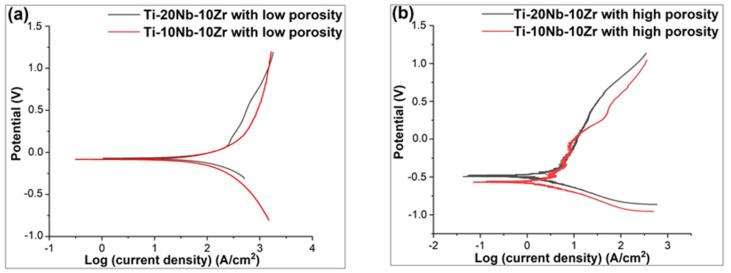
Results of electrochemical corrosion tests for Ti-xNb-10Zr (x: 10 and 20; at. %). (**a**) Alloys with low porosities; (**b**) Alloys with high porosities.

**Figure 10 materials-16-04240-f010:**
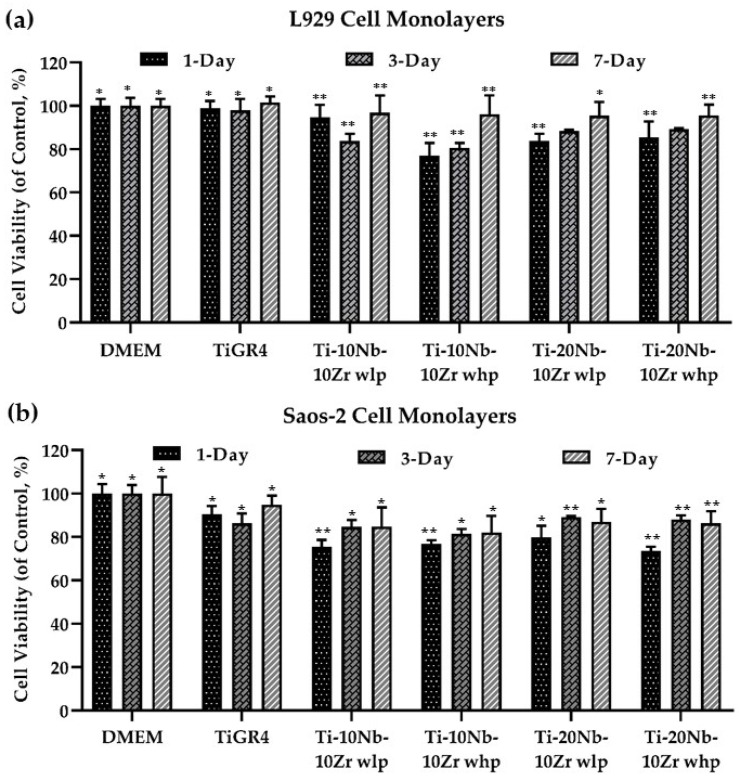
(**a**) L929 and (**b**) Saos-2 cell viability results for all Ti-(x)Nb-10Zr (x: 10 and 20; at. %) alloys with low porosity (wlp), high porosity (whp), and the reference TiGR4 extracts for 1-day, 3-day, and 7-day measured by MTT assay. Data represent mean ± SD, n = 3. *^,^ ** for *p* < 0.05.

**Figure 11 materials-16-04240-f011:**
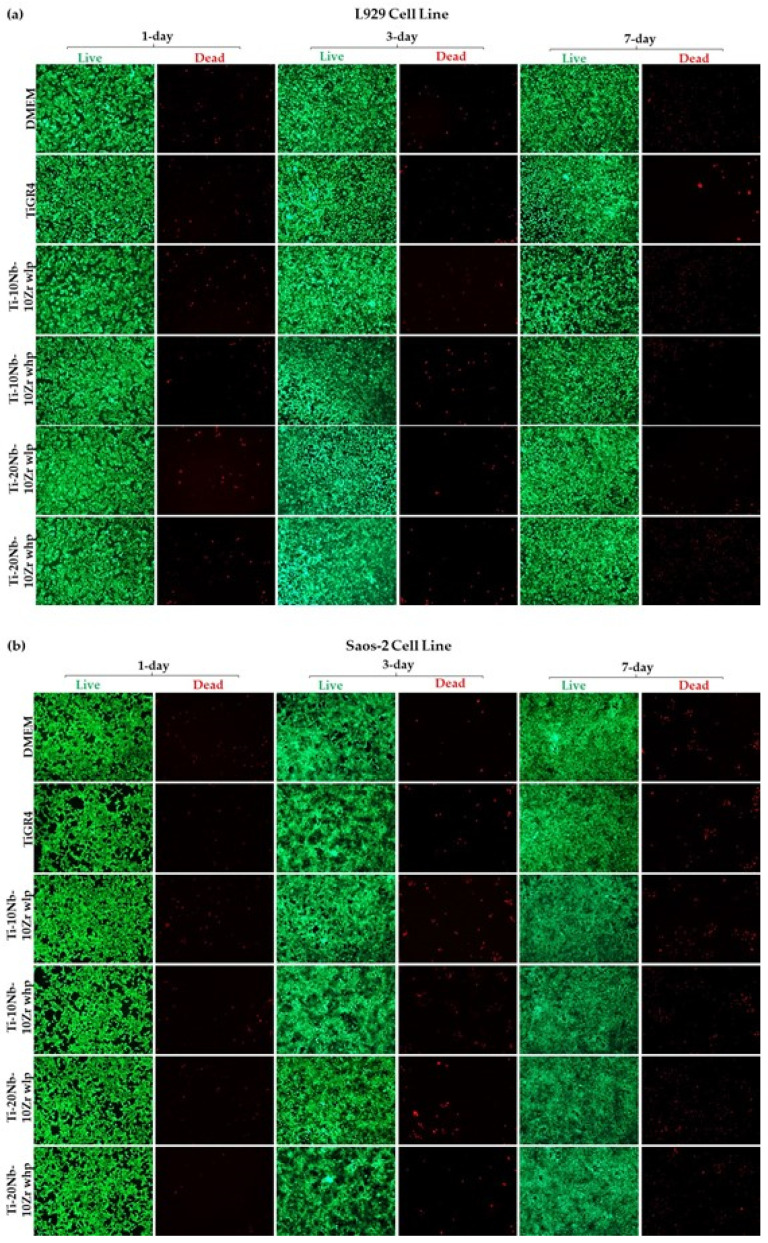
The images of (**a**) L929 and (**b**) Saos-2 cells upon exposure to Ti-xNb-10Zr (10, and 20; at. %) alloys with low porosity (wlp), with high porosity (whp), and the reference TiGR4 for 1-day, 3-day, and 7-day were taken by fluorescence microscope with 10× magnification.

**Figure 12 materials-16-04240-f012:**
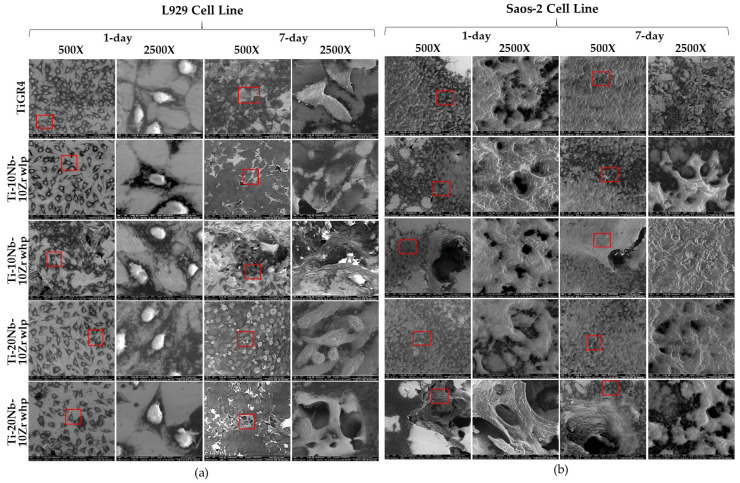
SEM images of viable (**a**) L929 and (**b**) Saos-2 cell lines for Ti-xNb-10Zr (x: 10 and 20; at. %) alloys with low porosity (wlp), with high porosity (whp), and the reference TiGR4 extracts for 1-day and 7-day. SEM images were taken with 500× magnification; 2500× magnification images taken from the red squares area in the 500× images.

**Figure 13 materials-16-04240-f013:**
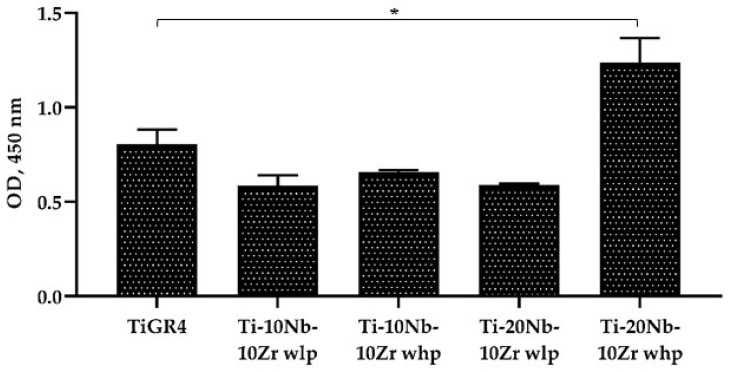
Adsorption of Fibronectin on Ti-xNb-10Zr (x: 10 and 20; at. %) with low porosity (wlp), with high porosity (whp), and the reference TiGR4 determined after 2 h incubation at 37 °C in a 5% CO_2_ atmosphere by ELISA method. Data represent mean ± SD, n = 3, * for *p* < 0.05.

**Table 1 materials-16-04240-t001:** Compositions of all Ti-xNb-10Zr (x: 10 and 20; at. %) alloys having differing porosities and process parameters used in this study (SH: Space holder).

Composition (at.%)	Mixing (h)	Compaction (MPa)	Temperature (°C)	Time (h)
Ti-10Nb-10Zr	10	300	1200	6
Ti-20Nb-10Zr	10	300	1200	6
Ti-10Nb-10Zr + 20SH (wt.%)	10	300	1200	6
Ti-20Nb-10Zr + 20SH (wt.%)	10	300	1200	6

**Table 2 materials-16-04240-t002:** The atomic compositions for individual EDS points.

Type of Alloy	EDS Points	Atomic Composition
Ti-10Nb-10Zr with low porosity	Point 1	Ti-30.83Nb-7.46Zr
	Point 2	Ti-11.37Nb-8.45Zr
	Point 3	Ti-12.17Nb-8.53Zr
Ti-20Nb-10Zr with low porosity	Point 1	Ti-97.4Nb-2.29Zr
	Point 2	Ti-18.85Nb-7.41Zr
	Point 3	Ti-11.67Nb-7.31Zr

**Table 3 materials-16-04240-t003:** The corrosion kinetics parameters for low and highly porous ternary Ti-xNb-10Zr (x: 10 and 20; at. %) alloys in Hank’s balanced salt solution.

	E_corr_ (mV.SCE)	β_a_ (V dec^−1^)	β_c_ (V dec^−1^)	i_corr_ (µA/cm^2^)	R_p_ (Ω/cm^2^)
Ti-10Nb-10Zr	−0.07	1.75	0.75	2.40	0.09
Ti-20Nb+10Zr	−0.077	1.52	0.4	2.26	0.06
Ti-10Nb-10Zr with low porosity	−0.55	0.86	0.24	0.67	0.12
Ti-20Nb+10Zr with high porosity	−0.48	0.92	0.18	0.39	0.16

## Data Availability

Both digital and non-digital data supporting this study are stored by the corresponding author at Brunel University and are available upon request.

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
