# Peer review of "Microstructural, Biomechanical, and In Vitro Studies of Ti-Nb-Zr Alloys Fabricated by Powder Metallurgy"

_materials, 2023, doi:10.3390/ma16124240_

Round 1
Reviewer 1 Report
The paper reports microstructure, compression tests, and in vitro studies of two Ti-Nb-Zr Alloys with different porosity levels. However, the results are not clarified in detail, and the analyses and discussions are incomplete (below comments). Therefore, I cannot recommend the paper for publication in the journal of Materials in its present form.
1. The abstract is incomplete. While the paper is focused on the effect of Nb, there is no information on the resultant microstructure, mechanical properties, etc.
2. How many compression tests were conducted for each sample condition? This should be mentioned in the experimental section. In addition, what are the error ranges of yield and tensile strengths reported in section 3.2?
3. There are several issues with Fig. 4. More importantly, there is no correlation between characterized microstructures through SEM and EBSD.
3.1. The color indexing of phase maps and the magnification of the image is misleading for the readers. Identical colors and scale bars (magnifications) should be presented for both alloys. What is the condition of the samples, i.e. low porosity or high porosity?
3.2. The identification of Nb in the EBSD results is not reliable. Is it really possible to distinguish Beta-Ti and Nb through EBSD? Please explain the procedure. The EBSD-software generated results have no meaning in this case.
3.3. Similar to the above comment: based on the SEM-EDS results, undissolved Nb only appears in the Ti-20Nb-10Zr alloy. Then, why Nb is detected in the EBSD results of the Ti-10Nb-10Zr alloy in Fig. 4c?
3.4. What is the average size of Nb particles detected by SEM? How about the EBSD results? What is the step size of EBSD scans shown in Fig.4? This should be mentioned in the experiments section too.
4. In Fig. 5b, both alloys show poor ductility. What are the properties requirements of such materials for biomedical applications? Also, how do the results compare with other alloys, e.g., Ti-6A-4V alloy?
5. The author measured the volume fraction of phases based on EBSD results, which is not the correct way in the present study. There are significant differences between the XRD and EBSD results.
6. Based on the XRD results of Fig. 1a and 1b, not only the Nb concentration but also the level of porosity is affecting the fraction of phases. What is the reason? In addition, what are the fraction of phases depending on alloy composition as well as porosity levels (i.e., 4 different sample conditions)?
7. Line 362 (page-10): “Nevertheless, all sintered alloys fabricated in this study 362 exhibited enhanced corrosion resistances, which was essential for implantology.” But, no comparison has been made to claim an enhanced corrosion resistance. In addition, how do the corrosion results compare with other alloys in the literature?
8. Line 385 (page 12): “Results revealed that increasing of Nb concentration from 10 (at. %) to 20 (at. %) in Ti-Zr mixtures resulted in a decrease in L929 cell viability and an improvement in Saos-2 cell viability.” Why Nb has such an effect?
9. Line 422 (page 13)” 10): “The results demonstrated that addition of SH agents and increasing of Nb concentration from 10 (at. %) to 20 (at. %) enhanced the fibronectin adsorption capacity.” Why Nb has such an effect?
Author Response
On behalf of all the corresponding authors, Yuksel Cetin and Yan Huang and the other authors Eyyup Murat Karakurt, Alper Incesu, Huseyin Demirtas, Mehmet, Kaya, Yasemin Yildizhan, Merve Tosun, We would like to appreciate the time and effort that you have provided valuable feedback on our manuscript entitled “Microstructural, Biomechanical and Biocompatibility Studies of Titanium-Niobium-Zirconium Alloys Fabricated by Powder Metallurgy”. We are grateful for your insightful comments and the critics on our paper. We believe your remarks and suggestions helped our paper to become complete. We have been able to incorporate changes to reflect the suggestions as requested. The manuscript improved to the present level. We deeply appreciate for your contribution. We hope that this version of the manuscript reached at the satisfactory level and acceptable for publication.
Responses to your Comments and Suggestions
The paper reports microstructure, compression tests, and in vitro studies of two Ti-Nb-Zr Alloys with different porosity levels. However, the results are not clarified in detail, and the analyses and discussions are incomplete (below comments). Therefore, I cannot recommend the paper for publication in the journal of Materials in its present form.
- The abstract is incomplete. While the paper is focused on the effect of Nb, there is no information on the resultant microstructure, mechanical properties, etc.
Response 1. The information on the resultant microstructure and mechanical properties was explained in the abstract as required.
Line 25-26 (page 1). Experimental results showed that the alloys had a dual phase microstructure composed by finely dispersed acicular hcp α-Ti needles in the bcc β-Ti matrix.
Line 29-30 (page 1). Noted that adding space holder agent played a more critical role on the mechanical behaviours of the alloys compared to adding niobium.
- How many compression tests were conducted for each sample condition? This should be mentioned in the experimental section. In addition, what are the error ranges of yield and tensile strengths reported in section 3.2?
Response 2. The number of compression tests and the error ranges of yield and tensile strengths were given in the related sections as required.
Line 153-157 (page 4-5). The compression tests were carried out 3 times for each sample condition on a universal tensile-compression testing machine (Zwick/Roell Z600) with a capacity of 100 kN, following EN 24506 standard. The average yield strengths and ultimate compressive strengths were determined from stress–strain curves (error range within 10%).
- There are several issues with Fig. 4. More importantly, there is no correlation between characterized microstructures through SEM and EBSD.
3.1. The color indexing of phase maps and the magnification of the image is misleading for the readers. Identical colors and scale bars (magnifications) should be presented for both alloys. What is the condition of the samples, i.e. low porosity or high porosity?
Response 3. The details of the relation between characterized microstructures through SEM and EBSD was explained below for each comment as required.
Response 3.1. EBSD map was modified according to the comment 3.1. Also, updated sample condition was written on the Figure 7 caption which was Figure 4 before revision.
Line 374-376 (page 11). Figure 7. EBSD data showing the microstructure (IQ map), grain morphology (IPF map) and phase constituents and distribution (Phase map) for both Ti-10Nb-10Zr alloy with low porosities: a) IQ map, b) IPF map, and c) Phase map, and for Ti-20Nb-10Zr alloy with high porosities: d) IQ map, e) IPF map, and f) Phase map.
3.2. The identification of Nb in the EBSD results is not reliable. Is it really possible to distinguish Beta-Ti and Nb through EBSD? Please explain the procedure. The EBSD-software generated results have no meaning in this case.
Response 3.2. The procedure was explained as required.
Line 365-368 (page 10). Lastly, undissolved Nb particles was surrounded by β phase. Therefore, it was difficult to identify the undissolved Nb cores from β phase on EBSD map. However, EBSD map revealed that the proportion of primary Nb phase in Ti-10Nb-10Zr alloy was smaller than that in Ti-20Nb-10Zr alloy, which were 12 % and 25 %, respectively.
3.3. Similar to the above comment: based on the SEM-EDS results, undissolved Nb only appears in the Ti-20Nb-10Zr alloy. Then, why Nb is detected in the EBSD results of the Ti-10Nb-10Zr alloy in Fig. 4c?
Response 3.3. The all samples had inhomogeneous structure as mentioned in the text, according to this concept, SEM image of the Tİ-10N-10Zr sample showed in the Figure 6 was updated. At this point, Undissolved Nb cores could be seen in all SEM images.
Line 338-340 (page 9). Accordingly, the existence of Nb cores in the microstructures was an indication of insufficient sintering temperature.
3.4. What is the average size of Nb particles detected by SEM? How about the EBSD results? What is the step size of EBSD scans shown in Fig.4? This should be mentioned in the experiments section too.
Response 3.4. All parameters of EBSD scans were shown in the text. In addition, the information about size of pure Nb cores was given in the Methods section 2.2..
Line 130-134 (page 4). Electron backscattered diffraction (EBSD) analysis was done by the Zeiss Supra 35VP fitted with high sensitivity Digi View camera (EDAX Inc., NJ, U.S.). The working distance, the accelerating voltage and condenser aperture employed were 12 mm, 20 kV and 120μm, respectively, in high current mode. Also, the step size selected was range of 0.2 μm -1 μm according to the size of area and grain size.
Line 365-366 (page 10). Undissolved Nb particles was surrounded by β phase. Therefore, it was difficult to measure the size of the undissolved Nb cores.
- In Fig. 5b, both alloys show poor ductility. What are the properties requirements of such materials for biomedical applications? Also, how do the results compare with other alloys, e.g., Ti-6A-4V alloy?
Response 4. Mechanical requirements of the implant material were given. Also, the results werecompared to Tİ-6Al-4V that is the most preferred implant material.
Line 46-47 (page 2). The mechanical properties of the implant material should be close to those of the natural bone .
Line 394-402 (page 11). Such behavior allows the implant to have a closer match to the mechanical properties of natural bone, minimizing stress shielding and facilitating improved load transfer between the implant and the bone tissue. When comparing the mechanical properties of stiff Ti-6Al-4V alloy with the sintered alloys achieved in this study, it can be said that the sintered alloys with low and high porosities can provide better mechanical compatibility with natural bone due to their reduced mechanical performances, light weight, and potentially enhanced biocompatibility. These findings revealed that the sintered alloys with low and high porosities could be a favorable candidate biomaterial for implantology.
- The author measured the volume fraction of phases based on EBSD results, which is not the correct way in the present study. There are significant differences between the XRD and EBSD results.
Response 5. Insufficient sintering regime, inhomogeneous structures were formed on the different surfaces of the samples. Therefore, XRD results and EBSD result were not identical. Normally, one analysis (only EBSD or XRD) was sufficient to show which phase formed at the surface of the sample. However, phase constituents determined by XRD and EBSD map analysis were consistent with each other.
- Based on the XRD results of Fig. 1a and 1b, not only the Nb concentration but also the level of porosity is affecting the fraction of phases. What is the reason? In addition, what are the fraction of phases depending on alloy composition as well as porosity levels (i.e., 4 different sample conditions)?
Response 6. The required information for the comment 6th based on XRD results was explained below.
Line 281-286 (page 7). In addition, alloying with Nb (strong β stabilizer) and zirconium (weak β stabilizer or neutral element) reduced the proportion of α phase in porous Ti-20Nb-10Zr (at. %) alloys with low and high porosities. Whereas β phase proportion increased by adding Nb, as Nb acted β-Ti stabilizer in the microstructure. On the other hand, adding space holder decreased the intensities of solute diffraction peaks of the both phases due to increase in diffusion pathways between the particles.
- Line 362 (page-10): “Nevertheless, all sintered alloys fabricated in this study 362 exhibited enhanced corrosion resistances, which was essential for implantology.” But, no comparison has been made to claim an enhanced corrosion resistance. In addition, how do the corrosion results compare with other alloys in the literature?
Response 7. All corrosion resistance results was modified to compare with each other.
Line 417-431 (page 12). The typical cyclic potentiodynamic polarization (Tafel) curves for Ti-xNb-10Zr alloys with low and high porosities were given in Figure 9. The summary of the results was also given in Table 3. Ecorr for Ti-xNb-10Zr (x: 10 and 20; at. %) alloys with low porosities were found in a range –0.07 mV to -0.077 mV, respectively, in Hank's balanced salt solution. At this point, Tafel curve of Ti-20Nb-10Zr alloy with low porosity was slightly closer to positive side of the corrosion potential, which revealed that this type of alloy had slightly lower current density (icorr: 2.26 µA.cm-2) compared to Ti-10Nb-10Zr alloy with low porosity (icorr: 2.40 µA.cm-2). As understood from these findings, increase Nb concentration from 10 to 20 slightly enhanced the corrosion resistance of Ti-20Nb-10Zr alloy with low porosity. Same tendency was observed in Ti-20Nb-10Zr with high porosity. This behaviour showed that adding niobium and zirconium was attributed to formation of oxide film. The potential caused an increase in anodic current because the increase in oxide thickness during testing was not sufficient to compensate for the potential increase [14]. As a result, the alloys studied in this study were a potential candidate for biomedical application.
- Line 385 (page 12): “Results revealed that increasing of Nb concentration from 10 (at. %) to 20 (at. %) in Ti-Zr mixtures resulted in a decrease in L929 cell viability and an improvement in Saos-2 cell viability.” Why Nb has such an effect?
Response 8.
Line 477-491 (page 16-17). The treatment of Ti-10Nb-10Zr alloy with high porosity compared with low porosity one was decreased the cell viability of L929 cell line on the 1-day and 3-day but not on the 7-day application. This could be due to adaptation of the cells to the conditions occurred with high porosity alloy and then the cell viability reached the same level resulted in with low porosity. The increase in Nb concentration from 10 (at. %) to 20 (at. %) in Ti-Zr alloys resulted in no significant change in L929 cell viability among different period of the treatments. The increase in Nb concentration from 10 (at. %) to 20 (at. %) in Ti-Zr alloys resulted in increase in Saos-2 cell viability within the different period of the treatments. It could be explained that the Saos-2 cell line might have sensitive metabolic activity to the Nb metals, which may induce the cell proliferation. In addition to these, Nb concentration might changed the surface characteristics of the alloys by forming NbxO films.
- Line 422 (page 13)” 10): “The results demonstrated that addition of SH agents and increasing of Nb concentration from 10 (at. %) to 20 (at. %) enhanced the fibronectin adsorption capacity.” Why Nb has such an effect?
Response 9.
Line 526-531 (page 17). Regarding biological characteristics of Nb, it is nontoxic and allergy-free metal, indicting acceptable biocompatibility and osteoconductivity. In this study, the increased level of Nb might induce much more mitochondrial activity, cell proliferation, fibroblast adsorption potential of biomaterial surfaces. In the previous work resulted ın similar founding’s that the comparative study on the biological performance of Nb, Ti, and stainless steel revealed that there were much more mitochondrial activity and cell proliferation on Nb compared to the others [35].
[35] Niinomi, M. Recent metallic materials for biomedical applications. Metall. Mater. Trans. A 2002, 33, 477

Reviewer 2 Report
Dear authors, your work is interesting, but the following changes must be made.
1. At the end of the introduction, it is necessary to describe in detail the goals of this work and the methods of analysis used.
2. In paragraph 2.1, it is necessary to add a graph of the heating and cooling of sintered materials and a schematic illustration of the mold.
3. Why was 20% SH agent added? Why not more or less?
4. Line 108 Need to add angles and scan speeds of XRD analysis.
5. Line 239 Each analyzed phase must be assigned a number from the ICDD library.
6. Line 271 It is necessary to indicate the sizes of macro and micro pores; the results of EDS are also given in the table.
7. P. 3.2 The results in Figure 6 do not correspond to the corrosion currents in the text. The polarization resistance is also incorrectly calculated because, based on the data obtained, you do not have a corrosion-resistant alloy but a superconductor. You must also indicate the cathodic and anodic slopes of the Tafel curves. Why did the corrosion potential decrease by almost 400 mV? The logarithmic scale must be correctly represented. All data must be presented in a table.
8. To clarify the corrosion characteristics of the alloy, it is necessary to make electrochemical impedance spectroscopy.
9. Discussion and conclusions are a statement of facts, which is usually given in technical reports. A deeper explanation of the results obtained is needed.
Author Response
Dear Reviewer,
On behalf of all the corresponding authors, Yuksel Cetin and Yan Huang and the other authors Eyyup Murat Karakurt, Alper Incesu, Huseyin Demirtas, Mehmet, Kaya, Yasemin Yildizhan, Merve Tosun, We would like to appreciate the time and effort that you have provided valuable feedback on our manuscript entitled “Microstructural, Biomechanical and Biocompatibility Studies of Titanium-Niobium-Zirconium Alloys Fabricated by Powder Metallurgy”. We are grateful for your insightful comments and the critics on our paper. We believe your remarks and suggestions helped our paper to become complete. We have been able to incorporate changes to reflect the suggestions as requested. The manuscript improved to the present level. We deeply appreciate for your contribution. We hope that this version of the manuscript reached at the satisfactory level and acceptable for publication.
Responses to your Comments and Suggestions
Dear authors, your work is interesting, but the following changes must be made.
- At the end of the introduction, it is necessary to describe in detail the goals of this work and the methods of analysis used.
Response 1. The aim of this study and the methods of analysis was given at the end of the introduction section as required.
Line 87-94 (page 2). This work aims to overcome drawbacks of the above-mentioned problems by producing Ti-xNb-Zr10 (x:10, and 20; at. %) alloys with differing porosities used as load-bearing implant that can mimic the bone structure. Microstructural analysis was performed by using various methods including scanning electron microscopy, energy dispersive spectroscopy, electron backscatter diffraction, and x-ray diffraction. Corrosion resistance was assessed via electrochemical polarisation tests, while mechanical behaviour was determined by uniaxial compressive tests. Biocompatibility was examined by performing MTT assay, fibronectin adsorption, and plasmid-DNA interaction assay.
- In paragraph 2.1, it is necessary to add a graph of the heating and cooling of sintered materials and a schematic illustration of the mold.
Response 2. Sintering regime graphs and schematic illustration of the mold were given as required.
Line 112-113 (page 3). Consequently, the graph of the sintering regimes for the sample alloys with low and high porosities is given in Figure 2.
Line 108-109 (page 3). Constructive shape of the steel die tool used is given in Figure 1.
- Why was 20% SH agent added? Why not more or less?
Response 3. The reason for adding a 20% spacer was explained as required.
Line 83-86 (page 2). Careful consideration and optimization of the space holder agent are required to achieve the desired mechanical properties in the final product. According to previous studies, the quantity of space holder agent used in this study has been determined as 20% (wt.).
- Line 108 Need to add angles and scan speeds of XRD analysis.
Response 4. The angles and scan speeds of XRD analysis were given as required.
Line 134-137 (page 4). The crystallographic characterizations of the sintered alloys with low and high porosities were performed by a Bruker D8 Advance Cu-Kα source. The step size, acquisition time, and the 2θ angle range used in this work were 0.02Ëš, 1s and 30Ëš- 80Ëš, respectively.
- Line 239 Each analysed phase must be assigned a number from the ICDD library.
Response 5. The analysed phases were assigned with numbers from the ICDD library.
Line 275-279 (page 7). X-ray diffraction spectra is presented in Figure 4. XRD results revealed the coexistence of both hcp α (ICDD PDF No. 00-044-1294) and bcc β (ICDD PDF No. 00-044-1288) peaks related to the Ti-based alloys as well as primary Nb phase, but at different volume fractions and no other obvious impurities peak such as oxide, hydride or intermetallic compound was detected.
- Line 271 It is necessary to indicate the sizes of macro and micro pores; the results of EDS are also given in the table.
Response 6. The mean sizes of macro and micro pores were given. Also, EDS results were given in the Table 2.
Line 319-323 (page 9). Average pore sizes of the sintered alloys increased from 67 μm to 117 μm for Ti-10Nb-10Zr and 82 μm to 143 μm for Ti-20Nb-10Zr, as a consequence of the solid-state sintering.
Line 313-314 (page 8). The atomic compositions for individual EDS points are listed in the Table 2.
Table 2. The atomic compositions for individual EDS points.
Type of alloy |
EDS points |
Atomic composition |
Ti-10Nb-10Zr with low porosity |
Point 1 |
Ti-30.83Nb-7.46Zr |
Point 2 |
Ti-11.37Nb-8.45Zr |
|
Point 3 |
Ti-12.17Nb-8.53Zr |
|
Ti-20Nb-10Zr with low porosity |
Point 1 |
Ti-97.4Nb-2.29Zr |
Point 2 |
Ti-18.85Nb-7.41Zr |
|
Point 3 |
Ti-11.67Nb-7.31Zr |
- 3.2 The results in Figure 6 do not correspond to the corrosion currents in the text. The polarization resistance is also incorrectly calculated because, based on the data obtained, you do not have a corrosion-resistant alloy but a superconductor. You must also indicate the cathodic and anodic slopes of the Tafel curves. Why did the corrosion potential decrease by almost 400 mV? The logarithmic scale must be correctly represented. All data must be presented in a table.
Response 7. All Tafel curves were modified. The summary of the results was also given in Table 3.
Line 417-431 (page 12).. The typical cyclic potentiodynamic polarization curves for Ti-xNb-10Zr alloys with low and high porosities are given in Figure 9. The summary of the results is also given in Table 3. Ecorr for Ti-xNb-10Zr (x: 10 and 20; at. %) alloys with low porosities were found in a range –0.07 mV to -0.077 mV, respectively, in Hank's balanced salt solution. At this point, Tafel curve of Ti-20Nb-10Zr alloy with low porosity was slightly closer to positive side of the corrosion potential, which revealed that this type of alloy had slightly lower current density (icorr: 2.26 µA.cm-2) compared to Ti-10Nb-10Zr alloy with low porosity (icorr: 2.40 µA.cm-2). As understood from these findings, increase Nb concentration from 10 to 20 slightly enhanced the corrosion resistance of Ti-20Nb-10Zr alloy with low porosity. Same tendency was observed in Ti-20Nb-10Zr with high porosity. This behaviour showed that adding niobium and zirconium was attributed to formation of oxide film. The potential caused an increase in anodic current because the increase in oxide thickness during testing was not sufficient to compensate for the potential increase [14]. As a result, the alloys studied in this thesis were a potential candidate for biomedical application.
- To clarify the corrosion characteristics of the alloy, it is necessary to make electrochemical impedance spectroscopy.
Response 8. Electrochemical impedance spectroscopy could not be performed.
- Discussion and conclusions are a statement of facts, which is usually given in technical reports. A deeper explanation of the results obtained is needed.
Response 9. All discussion was modified as requested in a way of deeper explanation of the results.
Line 568-618 (page 18-19). This result was expected and suggested that the use of NH4HCO3 as space holder agent was effective in generating high porosities. No residues of NH4HCO3 were found in the alloys studied after sintering stage, which indicated that the space holder agent was completely degraded to ammonia (NH3(g)) and carbon-dioxide (CO2(g)), leaving pores in the material [35, 36]. Accordingly, the alloys achieved in this study were fabricated by powder metallurgy with two categories of porosities, i.e., 20%–25% and 50%-56%, respectively. It is reasonable to believe that a full range of porosities required for clinical applications can be achieved by adjusting space holder agent additions and sintering conditions.
In implantology, the most critical point regarding a porous implant material is the size and distribution of macro and micro pores in microstructure. We can see conflicting research on the optimal pore size for implantology in the literature. However, most studies have showed that the optimal pore size should be in the range of 100 µm to 600 µm [36]. It is obvious that the pores within implant material should be larger than the minimum pore size allowing the blood and nutrient flow for bone ingrowth. Based on the results from the general porosity evaluations showed that alloys could be divided into low and highly porous categories, with pore size from 67 µm to 82 µm and 117 µm to 143 µm, respectively. Further, pore size and distribution found in all-porous alloys fabricated in this thesis were congruent with the pore size and distribution aimed for when the process began. However, porosity level of a load bearing implant materials can vary according to where they are used in the body. Therefore, it is quite difficult to say which alloy has the optimal porosity level. Noted that all-porosity levels achieved in this study were appropriate for mimicking the bone structure, which positively affected the biocompatibility properties of the alloys. The outcome of this study showed that the alloys obtained could be considered as a promising candidate for orthopedic applications.
In addition, the porosity characteristics could be effectively adjusted by adding SH agent Meanwhile, adding Nb concentration into Ti-Zr mixtures changed α/β ratio in the microstructure and formation of β, which is of vital importance for biomedical applications due to unique mechanical properties, could be increased despite of insufficient sintering conditions. Thus, phase constituents were able to be modified to achieve appropriate microstructures.
Potentiodynamic polarization results showed that the Ti-xNb-10Zr alloys with low porosities met the corrosion performance criteria as required for orthopaedic biomaterial use although these alloys had inhomogeneous structure caused by powder metallurgy. This is mainly due to the individual elements making up the alloys were corrosion resistant [40]. Similarly, same tendency was observed in the alloys with high porosities.
Previous studies revealed general porosities obtained were appropriate for reducing the risk of stress shielding effect [37, 38]. The alloys with high porosities exhibited mechanical performances closer to human bone than the alloys with low porosities. Such condition showed that powder metallurgy with space holder technique was crucial factor for adjusting the mechanical properties of the alloys studied in this study by generating extra porosity inside the microstructure.
Porosities obtained in this study provided adequate sites for bone ingrowth and cell proliferation [39]. However, SEM micrographs showing the morphology of the cells showed that better cell adhesion was observed in alloys with high porosities due to the interconnected open pore structures. In addition, Nb and Zr used as alloyant element in Ti matrix did not have any toxic or allergic effects on the cell viabilities. Such condition revealed that Al and V free Ti-based alloys could be fabricated by powder metallurgy method combined with space holder technique [40,41]. This work was able to minimize the above-mentioned problems by producing Ti-xNb-10Zr (x:10, and 20; at. %) alloys with different porosities used as load-bearing implant that can mimic the bone structure.

Round 2
Reviewer 1 Report
The revision is okay. Please consider the following comment before publication.
In Figure 7 (previously 4), the total fraction of phases is more than 100% (101% in both samples), which should be corrected.
Author Response
Dear Reviewer, On behalf of all the corresponding authors, Yuksel Cetin and Yan Huang and the other authors Eyyup Murat Karakurt, Alper Incesu, Huseyin Demirtas, Mehmet, Kaya, Yasemin Yildizhan, Merve Tosun, We would like to appreciate the time and effort that you have provided valuable feedback on our manuscript entitled “Microstructural, Biomechanical and Biocompatibility Studies of Titanium-Niobium-Zirconium Alloys Fabricated by Powder Metallurgy”. We are grateful for your insightful comments and the critics helped our paper to become complete. We deeply appreciate for your contribution and acceptance of our point-by-point response given to your comments and concerns. We hope that this version of the manuscript reached at the satisfactory level and acceptable for publication.
Comments and Suggestions
The revision is okay. Please consider the following comment before publication.
In Figure 7 (previously 4), the total fraction of phases is more than 100% (101% in both samples), which should be corrected.
Response: Page 11. In Figure 7 (previously 4), the total fraction of phases for both samples was corrected as 100% as required (Sample 1. Titanium (Alpha) 78%, Titanium (Beta) 11%, and Niobium 11%; Sample 2. Titanium (Alpha) 52%, Titanium (Beta) 23%, and Niobium 25%)
The revisions within the manuscript are visible with track changes option of word file.
I look forward to hearing your editorial comments and final decision.
Yours sincerely,

Reviewer 2 Report
G
Author Response
Dear Reviewer, On behalf of all the corresponding authors, Yuksel Cetin and Yan Huang and the other authors Eyyup Murat Karakurt, Alper Incesu, Huseyin Demirtas, Mehmet, Kaya, Yasemin Yildizhan, Merve Tosun, We would like to appreciate the time and effort that you have provided valuable feedback on our manuscript entitled “Microstructural, Biomechanical and Biocompatibility Studies of Titanium-Niobium-Zirconium Alloys Fabricated by Powder Metallurgy”. We are grateful for your insightful comments and the critics helped our paper to become complete. We deeply appreciate for your contribution and acceptance of our point-by-point response given to your comments and concerns. We hope that this version of the manuscript reached at the satisfactory level and acceptable for publication.
I look forward to hearing your comments and final decision.
Yours sincerely,
